# Therapeutic Potential of Ramalin Derivatives with Enhanced Stability in the Treatment of Alzheimer’s Disease

**DOI:** 10.3390/molecules29225223

**Published:** 2024-11-05

**Authors:** Tai Kyoung Kim, Ju-Mi Hong, Jaewon Kim, Kyung Hee Kim, Se Jong Han, Il-Chan Kim, Hyuncheol Oh, Dong-Gyu Jo, Joung Han Yim

**Affiliations:** 1Division of Polar Life Sciences, Korea Polar Research Institute, Incheon 21990, Republic of Korea; tkkim@kopri.re.kr (T.K.K.); wnal5555@kopri.re.kr (J.-M.H.); ashcercle@kopri.re.kr (J.K.); kh313@kopri.re.kr (K.H.K.); hansj@kopri.re.kr (S.J.H.); ickim@kopri.re.kr (I.-C.K.); 2Department of Plant Biotechnology, Korea University, Seoul 02841, Republic of Korea; 3Department of Chemistry, Hanseo University, Seosan 31962, Republic of Korea; 4College of Pharmacy, Wonkwang University, Iksan 54538, Republic of Korea; hoh@wku.ac.kr; 5School of Pharmacy, Sungkyunkwan University, Suwon 16419, Republic of Korea

**Keywords:** Alzheimer’s disease, Ramalin, derivatives, therapeutic potential, antioxidant, tau protein, β-secretase, anti-inflammatory, Ames

## Abstract

Alzheimer’s disease (AD) remains a significant public health challenge with limited effective treatment options. Ramalin, a compound derived from Antarctic lichens, has shown potential in the treatment of AD because of its strong antioxidant and anti-inflammatory properties. However, its instability and toxicity have hindered the development of Ramalin as a viable therapeutic agent. The primary objective of this study was to synthesize and evaluate novel Ramalin derivatives with enhanced stabilities and reduced toxic profiles, with the aim of retaining or improving their therapeutic potential against AD. The antioxidant, anti-inflammatory, anti-BACE-1, and anti-tau activities of four synthesized Ramalin derivatives (i.e., RA-Hyd-Me, RA-Hyd-Me-Tol, RA-Sali, and RA-Benzo) were evaluated. These derivatives demonstrated significantly improved stabilities compared to the parent compound, with RA-Sali giving the most promising results. More specifically, RA-Sali exhibited a potent BACE-1 inhibitory activity and effectively reduced tau phosphorylation, a critical factor in AD pathology. Despite exhibiting reduced antioxidant activities compared to the parent compound, these derivatives represent a potential multi-targeted approach for AD treatment, marking a significant step forward in the development of stable and effective AD therapeutics.

## 1. Introduction

Alzheimer’s disease (AD) remains an incurable condition despite significant global research efforts and investments aimed at developing effective treatments. In the United States, approximately 10% of the elderly population is affected by AD, and this number is projected to double to approximately 15 million by 2050 [1]. The public health impact of AD can be significantly mitigated through interventions aimed at preventing, halting, or slowing the disease; however, current medications have limited efficacies in reducing disease severity or slowing disease progression [2]. Such challenges in developing effective treatments for AD may stem from the narrow focus on disease treatment. In addition, the failure to develop treatments for dementia as a broader category presents another significant challenge [3]. Despite substantial investment to date, only seven treatments for AD have been approved by the Food and Drug Administration (FDA) [4,5]. These medications include cholinesterase inhibitors, NMDA receptor antagonists, and anti-amyloid beta (Aβ) monoclonal antibodies. Among these, five drugs are used to address the symptoms of AD, namely donepezil, rivastigmine, galantamine, memantine, and aducanumab (Aduhelm). Recently, the FDA accelerated the approval of lecanemab (Lequembi) for the treatment of early-stage AD owing to its validated efficacy [6]. Additionally, the beta-saccharide 1 (BACE1) inhibitor donanemab has been shown to slow the progression of cognitive decline by approximately 25% in early-stage patients [7]. However, these antibody-based treatments have not been effective in patients with advanced AD, and face numerous challenges, such as high costs and undesirable side effects [8,9].

Previous research has identified the potential of Ramalin, derived from the Antarctic lichen, as an anti-Alzheimer’s treatment owing to its strong antioxidant and anti-inflammatory properties [10]. Initial experiments showed promise; however, Ramalin was ultimately ruled out as a viable treatment because of its instability and toxicity, including cytotoxicity, splenomegaly, and genotoxicity. To address these issues, various Ramalin derivatives have been prepared to reduce its toxicity and enhance its physical characteristics [11]. For example, it was proposed that its stability could be improved by introducing a methyl group to hydrazine- or benzamido-type derivatives, creating a more stable structure by preventing oxidation (Figure 1). The structural instability of Ramalin is presumed to be caused by oxidation through its reaction with oxygen. This is evidenced by the increased stability when dissolved in deoxygenated water [12] compared to water containing dissolved oxygen. After dissolving Ramalin in water and deoxygenated water at a concentration of 10 mM and leaving it for 48 h at 25 °C, it was found that only about 53% of Ramalin remained in regular water, whereas 92% remained in deoxygenated water. Based on the mass product ion scan results (Appendix A), as shown in Figure 2, it is expected that Ramalin decomposes into the RA-Azo form. Although attempts were made to isolate and purify RA-Azo, it was found to decompose too rapidly, making isolation and purification impossible. In an effort to prevent the oxidation of Ramalin into the RA-Azo form, we designed derivatives to improve stability. We attempted modifications by introducing a methyl group into phenyl hydrazine to prevent conversion into the azo form or by linking the phenyl group to a benzophenone structure to block electron flow and mitigate oxidation, thereby enhancing stability.

In addition, it has been demonstrated that the structural instability resulting from the resonance between the hydrazine and phenol groups can vary depending on the phenyl functional group. Indeed, this was verified through the preparation of RA-PF derivatives, where the stability was enhanced by replacing the protons at positions 2 and 6 of the phenyl ring with fluoride [10]. However, although the stability was found to increase based on the type and position of the functional group [13], the obtained derivatives could not be unequivocally labeled as stable substances, because their stabilities were merely enhanced.

Antioxidant effects play a crucial role in the treatment of AD because oxidative stress is a key contributor to its pathogenesis [14,15,16]. Antioxidants mitigate neuronal damage and slow disease progression by neutralizing free radicals. In addition, anti-inflammatory effects are critically important in the context of AD, since chronic inflammation is known to exacerbate AD pathology by promoting the accumulation of amyloid plaques and tau tangles [17,18]. It is therefore desirable to prepare derivatives that exhibit desirable anti-inflammatory properties to both prevent and treat AD by reducing neuroinflammation.

The inhibition of BACE-1 is another significant therapeutic strategy for AD treatment, as this enzyme is responsible for the production of Aβ peptides that aggregate to form amyloid plaques, a hallmark of AD. Previous studies have demonstrated that BACE-1 inhibitors can reduce Aβ levels, thereby slowing or halting the progression of the disease [19]. Extensive research and development efforts in the area of BACE-1 inhibitors have led to the preparation of several nanomolar-range inhibitory compounds [20]. 

The inhibition of acetylcholinesterase (AChE) is also a well-established mechanism in AD treatment, as it increases the levels of acetylcholine in the brain, a neurotransmitter that is typically reduced in patients with AD [21]. Enhanced acetylcholine levels can improve cognitive function and alleviate the symptoms of dementia, thereby indicating the potential of AChE inhibitors to act as effective AD therapies [22,23].

Moreover, the tau inhibitory activity is also critical due to the ability of hyperphosphorylated tau proteins to form neurofibrillary tangles, another key pathological feature of AD [24]. By inhibiting tau phosphorylation, tangle formation could be inhibited, thereby offering an alternative approach for the treatment and prevention of AD [25,26,27].

Thus, in the current study, a range of derivatives are synthesized and tested for their ability to treat AD based on their antioxidant and anti-inflammatory effects, BACE-1 inhibitory activity, AChE inhibitory activity, and tau inhibitory activity. The primary objective of synthesizing these Ramalin derivatives is to evaluate whether the more stable derivatives exhibit comparable anti-AD activities to the parent compound, Ramalin. In addition, this study aims to address the stability and toxicity concerns associated with Ramalin, and the ultimate goal is to synthesize Ramalin derivatives that could potentially be developed into effective anti-AD drugs.

## 2. Results

### 2.1. Synthesis of the Ramalin Derivatives

The desired derivatives were prepared according to a previous method in the literature [10,28], with some modifications (i.e., modified reaction time and an increased reaction temperature of 0 °C). As outlined in Figure 1, this process employed 1-benzyl-N-Cbz-L-glutamic acid (p-Glu) as the starting material, which was activated in its anhydride form by the addition of triethylamine (TEA) and ethyl chloroformate (ECF) in dichloromethane (DCM). After allowing the reaction to proceed for 3 h, a TEA solution of the desired hydrazine was slowly introduced into the mixture, and stirring was continued at room temperature (RT, 25 °C) for an additional 16 h to synthesize p-Glu-Hyd-Me. Subsequently, the benzyl and benzyloxycarboxyl (Cbz) groups of p-Glu-Hyd-Me, which were purified by recrystallization from hexane and ethyl acetate (AcOEt), were subsequently removed by hydrogenation (Pd/C, H_2_) in methanol (MeOH) over ~16 h. The desired RA-Hyd-Me derivative was obtained via recrystallization from MeOH and EA.

The benzamido-type derivatives were synthesized using the same method. However, the deprotonated derivatives were poorly soluble in MeOH, leading to immediate crystallization, which prevented the removal of the Pd/C catalyst through filtration. Additionally, the low solubilities of these derivatives in water rendered purification more difficult, significantly reducing the yield. To address these solubility issues, an alternative approach based on the formation of Ramalin chloride derivatives was applied [28], generating the corresponding HCl salts. As illustrated in Figure 2, the coupling reaction was executed in a similar fashion by employing starting materials bearing tert-butoxyl carbonyl (Boc) and tert-butyl (t-Bu) groups instead of benzyl. The deprotection reaction was performed using 1 M HCl in AcOEt at RT (25 °C) for 16 h. Upon completion of the reaction, the HCl salts of the RA-BA derivatives were obtained through filtration and drying.

When dissolved in water at a concentration of 10 mM, Ramalin is highly unstable and poses challenges for drug development. As described above, the hydrazine component of Ramalin is mainly responsible for its instability; tuning of the phenyl group can partly address this issue. The incorporation of a methyl group in the form of 1-methyl-1-phenyl hydrazine, aimed at preventing the oxidation of hydrazine and its conversion into the azo form, significantly enhanced the stability of the material. Furthermore, augmentation of its stability was achieved by altering the electron flow between the hydrazine and phenyl groups through the introduction of a ketone group between these moieties. The stabilities of the prepared Ramalin derivatives were therefore similarly assessed in water (10 mM). For this purpose, high-performance liquid chromatography (HPLC) was employed to measure any changes in these species upon storage at 60 °C (Table 1). Notably, the four synthesized Ramalin derivatives (RA-Hyd-Me, RA-Hyd-Me-Tol, RA-Sali, RA-Benzo) were found to exhibit outstanding stabilities.

It was hypothesized that the synthesized Ramalin derivatives could possess the ability to traverse the blood–brain barrier (BBB) because of their low molecular weights [29]. In the early stages of AD, the BBB becomes permeable, potentially allowing the administration of small molecules weighing <400 Da [30,31]. Thus, to develop small-molecule AD treatments that can effectively penetrate the BBB, it is necessary to confirm appropriate molecular weights, and also to evaluate their hydrogen bond numbers [32]. More specifically, the lipid solubility of a drug is inversely correlated with the number of hydrogen bonds formed in polar solvents such as water, with the optimal number of hydrogen bonds being <8–10 [33,34,35]. Notably, the synthesized Ramalin derivatives exhibited negative AlogP values and low lipophilicities. Additionally, their molecular weights were <300 Da, and all derivatives exhibited calculated hydrogen bond donor (HBD) values of ≤5, along with hydrogen bond acceptor (HBA) values of ≤10 (Table 2).

### 2.2. Antioxidant Effects of the Ramalin Derivatives

The antioxidant activities of the prepared Ramalin derivatives were assessed using the 2,2-diphenyl-1-picrylhydrazine hydrate (DPPH) assay (Table 2). It was found that their antioxidant activities were significantly lower than that of Ramalin itself, consistent with previous studies performed using Ramalin derivatives [10,28]. This can be accounted for by considering that Ramalin and its derivatives possess connected hydrazine and phenol groups, resulting in an antioxidant effect similar to that of Ramalin. In addition, the RA-Hyd-Me and RA-Hyd-Me-Tol derivatives, bearing a methyl group connected to the hydrazine moiety, were anticipated to exhibit diminished antioxidant effects owing to the disruption of the electron flow between the phenyl and hydrazine groups, thereby hindering the formation of azo-type double bonds. This is reflected in their DPPH IC_50_ values, with RA-Hyd-Me and RA-Hyd-Me-Tol giving IC_50_ values of 14.38 and 12.37 µM, respectively. This impact was more pronounced in the cases of the RA-Benzo and RA-Sali derivatives, where the presence of a ketone group between the phenyl group and the hydrazine moiety led to complete blockage of the electron flow. As a result, these derivatives exhibited significantly reduced antioxidant effects, with RA-Sali and RA-Benzo showing IC_50_ values of 1052 and 3860 µM, respectively. Thus, the described structural modifications led to enhanced stabilities for the derivatives compared to Ramalin, leading to significantly reduced antioxidant effects. Although the antioxidant effect may contribute to the anti-AD effects, the poor stability and toxicity resulting from an enhanced antioxidant effect were deemed problematic in the context of drug development. Consequently, the derivatives were designed with a primary emphasis on maximizing stability. 

### 2.3. BACE-1 Inhibitory Activities of the Ramalin Derivatives

Previously, it has been reported that Ramalin exhibited an IC_50_ value for BACE-1 inhibition of 17.66 ± 2.74 µM [10]. In the current study, Panvera^®^’s BACE-1 Fluorescence Resonance Energy Transfer (FRET) assay kit (P2985, PanVera Corp., Madison, WI, USA) was employed to assess the BACE-1 inhibitory activities of the Ramalin derivatives [10,36], and the obtained IC_50_ values are presented in Table 2. Commercially available LY2811376 (Eli Lilly, Indianapolis, IN, USA) was employed as the standard positive control, and all experiments were conducted in triplicate. 

All Ramalin derivatives displayed IC_50_ values in the micromolar range, with their BACE-1 inhibitory activities varying based on structural differences (Table 3). For RA-Hyd-Me, the presence of a methyl group attached to the hydrazine moiety led to an improved IC_50_ value of 15.5 ± 5.8 µM, representing a slightly stronger inhibitory activity than Ramalin. In contrast, RA-Hyd-Me-Tol showed a slightly reduced inhibitory activity with an IC_50_ value of 22.49 ± 1.76 µM. RA-Sali, containing a salicylamide group, exhibited the most potent activity, with an IC_50_ value of 8.44 ± 5.16 µM, potentially due to the hydroxyl group’s effective interaction with the BACE-1 active site. In contrast, RA-Benzo, which features a benzamido group without any additional hydroxyl group, exhibited a significantly increased IC_50_ value of 143.51 ± 87.12 µM, indicating reduced inhibition of BACE-1. These results suggest that functional groups, such as the hydroxyl group in RA-Sali, play a key role in enhancing BACE-1 inhibitory activity, emphasizing the importance of specific structural features in the design of effective inhibitors. 

### 2.4. Determination of the P-Tau Levels Using ELISA

The levels of phosphorylated tau (P-tau) were measured using enzyme-linked immunosorbent assay (ELISA) kits after treating the cells with the various Ramalin derivatives. The four prepared derivatives were tested at concentrations of 20 µM, and their inhibitory effects against P-tau were quantified. As shown in Figure 3, RA-Sali demonstrated the most significant inhibitory activity, reducing the tau levels by ~40% at the tested concentrations. Considering previous reports that Ramalin itself inhibited tau aggregation by 35.8% [28], the pronounced effect observed for RA-Sali can be attributed to its unique structural features, particularly the salicylamide group (containing hydroxyl and ketone groups), which likely enhances its enzyme binding affinity and interaction with the tau phosphorylation pathways. In contrast, the other three derivatives (RA-Hyd-Me, RA-Hyd-Me-Tol, and RA-Benzo) exhibited modest inhibitory activities ranging from 10 to 15%. Notably, RA-Hyd-Me and RA-Hyd-Me-Tol, both featuring a methyl group attached to the hydrazine moiety, displayed similar levels of tau inhibition, suggesting that methyl addition did not significantly enhance the activity. Furthermore, RA-benzo, containing a benzamido group, demonstrated the lowest inhibition among the four derivatives, likely due to the absence of additional functional groups that can facilitate stronger interactions with the tau protein. These results highlight the importance of specific functional groups and their positions within the molecular structure to enhance tau inhibition activity.

### 2.5. Cell Viability Assay

The toxicities of the prepared Ramalin derivatives against the RAW 264.7 cell line were subsequently assessed using the MTT assay. Furthermore, safety thresholds were established for the derivatives based on the percentages of viable cells. For this purpose, untreated cells were used as the control, representing a 100% survival rate. As outlined in Figure 4, all derivatives led to cell survival rates <80%, even at the highest concentration of 25 µM.

### 2.6. NO and NLRP3 Inhibition Activity Tests

The roles of anti-inflammatory agents in preventing and treating AD have been extensively investigated, indicating that Aβ generation induces inflammation, activating microglia and triggering the secretion of inflammatory cytokines, such as TNF-α, INF-γ, and interleukins [17,18]. The induced cytokines further contribute to AD pathology by hyperphosphorylation of the tau protein [19]. Additionally, the activation of NLRP3 has been implicated in the pathogenesis of AD [20,21], and this inflammatory response fosters the generation of Aβ and tau, ultimately contributing to disease progression. Thus, to determine the anti-inflammatory activities of the prepared derivatives, we measured the inhibition of nitrogen oxide (NO) and NLRP3 in LPS-induced RAW264.7 cells. The active concentration for activity assessment was established as 25 µM, at which cell viability had been previously confirmed. Consequently, the NO inhibitory activities of the derivatives were found to be ~20% (with the exception of the RA-Benzo derivative), while RA-Sali exhibited the highest NLRP3 inhibitory activity of ~25% at the test concentration (Figure 5).

### 2.7. In Vitro Microplate Mini Ames Assay

To evaluate whether the prepared derivatives could reduce the mutagenic potential of the parent compound, Ramalin, Ames tests were conducted using two bacterial strains, namely Salmonella typhimurium TA98 and TA100 [37,38], and the mutagenic potentials were evaluated using xenometric calculations. The bacterial strains were cultured at 37 °C for 12–16 h, and the test substances were serially diluted to six concentration levels. The bacterial suspensions were then exposed to the test substances, along with controls, in the presence or absence of a metabolic activation system (S9) at 37 °C for 90 min. After exposure, the mixtures were combined with the indicator medium, distributed into 384-well plates, and incubated at 37 °C for 24–48 h. According to the results listed in Table 4, Ramalin exhibited mutagenic activity under specific conditions, whereas none of the newly synthesized derivatives exhibited mutagenicity. More specifically, Ramalin exhibits a more pronounced mutagenic activity in the presence of the S9 metabolic activation system, giving positive results at concentrations ≥4.88 µg/mL for TA98 and TA100 both in the presence and absence of S9. In the presence of S9, Ramalin showed positive results which were observed at concentrations of ≥4.88 µg/mL for TA98 and ≥9.77 µg/mL for TA100. In contrast, the synthesized derivatives (RA-Hyd-Me, RA-Hyd-Me-Tol, RA-Sali, and RA-Benzo) did not exhibit mutagenic activity against either strain under any of the conditions tested, even in the presence of S9. This lack of mutagenicity suggests that the structural modifications introduced to generate these derivatives effectively mitigated the mutagenic potential of the parent compound, Ramalin.

### 2.8. Acetylcholinesterase Inhibitory Activity of Ramalin and Its Derivatives

The acetylcholinesterase (AChE) inhibitory activity of Ramalin and its derivatives was evaluated using a standard assay, and the results indicated that the inhibitory activity was generally minimal (Figure 6). Ramalin exhibited a 4.4% AChE inhibition at a concentration of 50 μM, while RA-Sali showed a 7.8% AChE inhibition at the same concentration. In contrast, the control inhibitor, donepezil, demonstrated significantly higher inhibitory activity, confirming the validity of the assay. These findings indicate that Ramalin and its derivatives have minimal AChE inhibitory activity, suggesting limited potential as AChE inhibitors.

## 3. Discussion

This study focused on the synthesis and evaluation of various Ramalin derivatives for the treatment of AD. Ramalin, which is derived from the Antarctic lichen, is known to demonstrate strong antioxidant and anti-inflammatory properties, thereby rendering it a promising candidate for AD treatment. However, its inherent instability and toxicity significantly limit its potential as a viable therapeutic agent. Addressing these limitations by developing more stable and less toxic derivatives was therefore the primary motivation for this study.

Thus, four novel Ramalin derivatives were prepared, namely RA-Hyd-Me, RA-Hyd-Me-Tol, RA-Sali, and RA-Benzo, and their chemical stabilities were extensively evaluated under various conditions. The results revealed that these derivatives maintained their structural integrity for extended periods, even at elevated temperatures. In contrast, the parent compound, Ramalin, exhibited significant degradation, undergoing rapid decomposition within a few hours at 60 °C. The incorporation of hydrazino methyl and benzamido functional groups therefore appeared to play a crucial role in enhancing its stability by preventing oxidative degradation and stabilizing the electron flow within the structure. These findings are particularly significant as they address one of the major challenges associated with AD drug development, namely the stability of the potential therapeutic agent. Notably, stable compounds are essential for the development of reliable and effective drugs because instability can lead to inconsistent therapeutic outcomes, reduced efficacies, and increased side effects. By improving the stability of Ramalin, this study has taken a crucial step towards developing more reliable AD treatments. 

The mutagenic potentials of the prepared derivatives were assessed using a mini Ames test with two bacterial strains, namely Salmonella typhimurium TA98 and TA100. The results indicated that unlike the parent compound Ramalin, which demonstrated significant mutagenic activity, particularly in the presence of the S9 metabolic activation system, the synthesized derivatives did not exhibit mutagenicity under the tested conditions. This lack of mutagenicity suggests that the structural modifications introduced into the derivatives effectively mitigated the mutagenic potential observed for Ramalin.

However, it should be noted that the above improvements come at the cost of reduced antioxidant activities, with the synthesized derivatives exhibiting lower antioxidant activities than Ramalin itself. These reductions were attributed to the structural modifications intended to inhibit oxidation by disrupting the electron flow between the hydrazine and phenyl groups. More specifically, RA-Benzo and RA-Sali exhibited diminished antioxidant effects, with their activities reduced to the mM level, while RA-Hyd-Me and RA-Hyd-Me-Tol maintained their antioxidant effect at concentrations of 12–14 µM. This tradeoff between stability and antioxidant activity underscores the complexity of drug development, wherein multiple factors must be balanced to achieve an optimal therapeutic profile. 

Despite their reduced antioxidant activities, the synthesized derivatives exhibited other promising therapeutic properties, such as improved BACE-1 inhibitory activities and anti-tau activities, which are critical in the context of AD. The abilities of RA-Hyd-Me and RA-Sali to inhibit BACE-1, an enzyme involved in the production of Aβ, suggest that these compounds could reduce the formation of amyloid plaques, a hallmark of AD pathology. In particular, RA-Sali demonstrated the most effective BACE-1 inhibitory activity, with an IC_50_ value lower than that of the positive control, LY2811376. The effects of the various derivatives on tau phosphorylation were also evaluated using the ELISA approach. The cellular levels of p-tau were measured after treatment with the Ramalin derivatives (20 µM), and it was found that among the derivatives, RA-Sali exhibited the most significant inhibitory activity with an inhibition level of ~40%, further confirming the potential of these derivatives for use as multi-target therapeutic agents in AD treatment. The remaining three derivatives (RA-Hyd-Me, RA-Hyd-Me-Tol, and RA-Benzo) showed inhibitory activities of approximately 10–15%. Despite these promising findings, neither Ramalin nor any of its four derivatives exhibited inhibitory activities against AChE. This suggests that although these compounds may have potential as multi-target therapeutic agents for AD, they may not be effective in targeting AChE, a common target in existing AD treatments.

This study also examined the potential of the derivatives to cross the BBB. Given that the BBB is a significant obstacle to drug delivery into the central nervous system, the low molecular weights and favorable physicochemical properties of these derivatives suggest that they may effectively traverse the BBB and directly exert their therapeutic effects in the brain. This hypothesis is supported by the calculated HBD and HBA values, as well as the derivative molecular weights, which are within the ranges typically associated with BBB permeability [39]. However, the passage of these derivatives through the BBB may still present challenges, particularly in the case of the RA-Sali derivative, which has a higher HBA value of 6, an HBD value of 5, and a polar surface area (PSA) of 141.75 Å^2^ [32]. Nevertheless, their low molecular weights provide a potential advantage for overcoming this limitation, especially when considering the possibility of enhancing the BBB permeability or performing further modifications to improve the physicochemical properties of Ramalin derivatives.

Thus, although the synthesized Ramalin derivatives show promise as potential therapeutic agents for AD, further optimization is necessary. Indeed, the current anti-AD effects are considered insufficient, and additional structural refinements are required to enhance their antioxidant, anti-inflammatory, anti-BACE-1, and anti-tau activities. Future studies should consider improvements in BBB permeability to ensure that these compounds can effectively reach their target in the brain. However, despite these challenges, this study represents a significant advancement in the development of stable and effective AD treatments and lays the groundwork for future therapeutic strategies.

## 4. Materials and Methods

### 4.1. General Experimental Information

All solvents and reagents were obtained from Merck (Darmstadt, Germany) or TCI (Tokyo, Japan) and used without further purification. All glassware was thoroughly washed and dried in a drying oven (60 °C) or flamed and cooled under a stream of dry argon prior to use. Filters were obtained from GE healthcare (GF/F, 0.7 µm, Whatman, UK). All reactions were performed under an inert argon atmosphere. Solvents and liquid reagents were transferred to a syringe prior to use. Organic extracts were dried over Na_2_SO_4_ and concentrated under reduced pressure in a rotary evaporator (Eyela, Tokyo, Japan). Residual solvent from the extracts was removed under a high vacuum (Vacuubrand RZ 2.5, Wertheim, Germany, 1 × 10^−2^ mbar). Purification was performed using a Yamazen Smart Flash EPCLC AI-580S (Yamazen, Osaka, Japan) medium-pressure liquid chromatography (MPLC) system. Accurate mass spectra were obtained using an AB Sciex Triple TOF 4600 (Framingham, MA, USA) instrument, with the interface in the direct injection mode. Infrared (IR) spectra were collected using a Bruker Vertex80V FT-IR spectrometer (Bruker, Billerica, MA, USA), equipped with a vacuum system. Nuclear magnetic resonance (NMR) spectra were obtained on a Jeol JNM ECP-400 spectrometer (Jeol Ltd., Tokyo, Japan) using a mixture of D_2_O (with 0.01 mg/mL sodium trimethylsilylpropanesulfonate (DSS))/acetone-d_6_ (6:1 *v*/*v*) or dimethyl sulfoxide (DMSO)-d_6_ as solvents. The internal reference or residual solvent signals were utilized for referencing (D_2_O (with DSS)/acetone-d_6_: dH 0.00/dC 29.8; DMSO-d_6_: dH 2.50/dC 39.5). The peak-splitting patterns were abbreviated as m, s, d, t, dd, ddd, and br for multiplets, singlets, doublets, triplets, doublets of doublets, doublets of doublets of doublets, and broad, respectively. Microplate (Thermo Scientific Inc., San Diego, CA, USA) and multimode plate readers (MultistkanTM GO, Thermo Scientific, Waltham, MA, USA) were used for absorbance analyses.

### 4.2. Synthesis and Characterization

#### 4.2.1. General Method for the Synthesis of p-Glu-Hyd-Me

(S)-5-(benzyloxy)-4-(((benzyloxy)carbonyl)amino)-5-oxopentanoic acid (2.0 g, 5.39 mmol) was dissolved in 50 mL of DCM, and the reaction mixture was cooled to 0 °C. TEA (1.2 eq, 6.47 mmol, 902 µL) was gradually added to the mixture, and after 10 min, ECF (1.2 eq, 6.47 mmol, 615 µL) was added dropwise to the mixture over the course of 1 h. The mixture was then stirred at 0 °C for 4 h. Separately, phenyl hydrazine (1.2 eq, 6.47 mmol) and TEA (1.2 eq, 6.47 mmol, 902 µL) were added to a 100 mL pear-shaped flask. This mixture was then added slowly to the main reaction flask over 1 h while maintaining the temperature at 0 °C. Once the addition of hydrazine was complete, the reaction mixture was allowed to warm to RT and stirred for an additional 16 h. Upon completion of the reaction, the organic layer was sequentially washed with distilled water, 1 N HCl, 0.5 N NaHCO_3_, and again with distilled water, before being separated and collected. The organic phase was dried over Na_2_SO_4_, followed by concentration. The desired product was purified via recrystallization using an AcOEt/n-hexane mixture (1:5).

#### 4.2.2. General Method for the Synthesis of RA-Hyd-Me Derivatives

The p-Glu-Hyd-Me analog (4.5 mmol) and palladium on carbon (10 wt%) were dissolved in MeOH (200 mL) and stirred under a hydrogen atmosphere (1 atm, using a hydrogen balloon) for 16 h. Once the reaction was complete, the mixture was passed through a 0.4 µm glass microfiber filter. The filtrate was then concentrated, followed by purification through recrystallization from a 1:5 mixture of methanol and AcOEt.

*N^5^-(methyl(phenyl)amino)-L-glutamine (RA-Hyd-Me)*. From (*S*)-5-(benzyloxy)-4-(((benzyloxy)carbonyl)amino)-5-oxopentanoic acid; 1.0 g, 81%, white solid; IR (ν cm^−1^, KBr): 3195, 3030, 2932, 2856, 1656, 1582, 1517, 1451, 1403, 1309, 1250; ^1^H NMR (400 MHz, D_2_O/acetone-d_6_ 6/1): δ 7.33 (m, 2H, PhH), 6.99 (m, 1H, PhH), 6.94 (m, 2H, PhH), 3.86 (t, *J* = 6.0 Hz, 1H, H-2), 3.09 (s, 3H, 1′-N-CH_3_), 2.61 (m, 2H, H-3), 2.26 (m, 2H, H-4); ^13^C NMR (100 MHz D_2_O/acetone-d_6_ 6/1): δ 174.1, 174.0, 149.3, 129.6, 120.9, 114.0, 54.3, 40.6, 29.7, 26.3; HRESIMS *m*/*z* 252.1354 [M + H]^+^ (calcd for C_12_H_18_N_3_O_3_, 252.1348).*N^5^-(methyl(m-tolyl)amino)-L-glutamine (RA-Hyd-Me-Tol)*. From (*S*)-5-(benzyloxy)-4-(((benzyloxy)carbonyl)amino)-5-oxopentanoic acid; 1.24 g, 87%, white solid; IR (ν cm^−1^, KBr): 3225, 3030, 2918, 2613, 1655, 1582, 1515, 1450, 1403, 1309, 1251; ^1^H NMR (400 MHz, D_2_O/acetone-d_6_ 6/2): δ 7.26 (t, *J* = 8.0 Hz, 1H, PhH), 6.85 (t, *J* = 7.2 Hz, 1H, PhH), 6.81 (s, 1H, PhH), 6.79 (m, 1H, PhH), 3.86 (t, *J* = 6.0 Hz, 1H, H-2), 3.11 (s, 3H, 1′-N-CH_3_), 2.61 (m, 2H, H-3), 2.33 (s, 3H, 3′-CH_3_), 2.26 (m, 2H, H-4); ^13^C NMR (100 MHz D_2_O/acetone-d_6_ 6/2): δ173.9, 174.0, 149.5, 140.0, 129.6, 121.6, 114.5, 111.2, 54.4, 40.7, 29.8, 26.2, 20.8; HRESIMS *m*/*z* 266.1513 [M + H]^+^ (calcd for C_13_H_20_N_3_O_3_, 266.1505).

#### 4.2.3. General Method for the Synthesis of Glu-Boc-RA

The starting material, (S)-5-(tert-butoxy)-4-((tert-butoxycarbonyl)amino)-5-oxopentanoic acid (2.0 g, 6.59 mmol), was dissolved in DCM (100 mL), and the temperature was reduced to 0 °C. Once the temperature was stable, TEA (1.2 eq, 7.91 mmol) was slowly introduced into the reaction and stirred for about 10 min. Then, ECF (1.2 eq, 7.91 mmol) was added dropwise over the course of an hour while keeping the temperature at 0 °C, followed by continuous stirring for 4 h. Separately, hydrazine HCl salt was dissolved in DCM (20 mL), followed by the addition of TEA (1.5 eq, 9.89 mmol). This solution was then slowly added to the main reaction mixture and stirred for approximately 10 min. After hydrazine addition, the temperature was allowed to rise to room temperature (approximately 24 °C), and the reaction was stirred for 16 h. Upon completion, the organic phase was washed with distilled water, 1 N HCl, 0.5 M NaHCO_3_, and distilled water again to separate the layers. The organic phase was then dried over MgSO_4_ and concentrated. Purification was performed using MPLC (C18 resin) with water and MeOH.

#### 4.2.4. General Method for the Synthesis of RA-BA Derivatives

An appropriate Boc-glu-Hyd analog (7.0 mmol) was dissolved in 1 M HCl in AcOEt (100 mL, 100 mmol). The reaction was allowed to proceed at room temperature (24 °C) for approximately 18 h. The resulting white solid was filtered and washed with AcOEt and n-hexane. The filtered white solid was dried under a vacuum to obtain the RA-BA derivative.

*N^5^-benzamido-L-glutamine hydrochloride (RA-Benzo)*. From (*S*)-5-(tert-butoxy)-4-((tert-butoxycarbonyl)amino)-5-oxopentanoic acid; 1.69 g, 85%, white solid; IR (ν cm^−1^, KBr): 3183, 2928, 1729, 1638, 1576, 1508, 1484, 1237; ^1^H NMR (400 MHz, DMSO-d_6_): δ 10.38 (s, 1H, 1′-NH), 10.09 (s, 1H, 5-NH), 8.49 (br s, 2H, 2-NH_2_) 7.86 (d, *J* = 7.3 Hz, 2H, PhH), 7.57 (t, *J* = 7.3 Hz, 1H, PhH), 7.50 (t, *J* = 7.3 Hz, 1H, PhH), 3.98 (t, *J* = 6.4 Hz, 1H, H-2), 2.52 (m, 2H, H-4), 2.08 (m, 3H, H-3); ^13^C NMR (100 MHz, DMSO-d_6_): δ170.7, 170.4, 165.5, 131.9, 128.5, 127.5, 132.4, 51.4, 28.9, 26.0; HRESIMS *m*/*z* 266.1139 [M + H]^+^ (calcd for C_12_H_16_N_3_O_4_, 266.1140).*N^5^-(2-hydroxybenzamido)-L-glutamine hydrochloride (RA-Sali)*. From (*S*)-5-(tert-butoxy)-4-((tert-butoxycarbonyl)amino)-5-oxopentanoic acid; 1.88 g, 90%, white solid; IR (ν cm^−1^, KBr): 2931, 1713, 1604, 1484, 1210; ^1^H NMR (400 MHz, DMSO-d_6_): δ 7.84 (dd, *J* = 1.7, 8.0 Hz, 1H, PhH), 7.51 (ddd, *J* = 1.7, 7.3, 8.7 Hz, 1H, PhH), 7.05 (m, 2H, PhH), 3.86 (t, *J* = 6.1 Hz, 1H, H-2), 2.61 (m, 2H, H-4), 2.26 (m, 3H, H-3); ^13^C NMR (100 MHz, DMSO-d_6_): δ173.8, 173.7, 168.6, 157.6, 135.3, 129.5, 120.7, 117.5, 115.2, 54.4, 29.7, 26.3; HRESIMS *m*/*z* 282.1087 [M + H]^+^ (calcd for C_13_H_20_N_3_O_3_, 282.1090).

### 4.3. DPPH Assay (In Vitro)

Following the method described by Blois et al. [40], the DPPH radical scavenging activity of Ramalin and its derivatives was evaluated. In summary, 150 µL of Ramalin, its derivatives, and butylated hydroxyanisole (BHA) at concentrations of 10, 5, 2.5, and 1 µM in methanol was combined with 50 µL of 0.1 mM DPPH in methanol. The mixture was then kept in the dark at RT for 30 min. Afterward, the absorbance was measured at 540 nm.

### 4.4. BACE-1 Inhibition Assay

A BACE-1 inhibition assay was performed using a β-Secretase FRET kit (BACE-1, Thermo Fisher Scientific, San Diego, CA, USA) following the manufacturer’s instructions. The procedure followed the previously described protocol. A stock solution of Ramalin and its derivatives was prepared in deionized distilled water (DDW) at a concentration of 20 mM. This stock was further diluted in assay buffer to achieve final concentrations of 50, 25, 12.5, 6.25, 3.12, 1.56, 0.78, 0.39, 0.2, and 0.1 µM in each well. In black 96-well microplates, 10 µL of the BACE-1 substrate was added to the diluted samples. The reaction was initiated by adding 10 µL of 3× BACE-1 enzyme to each well. The plates were incubated for 60 min at RT in the dark. After incubation, the reaction was stopped by adding 10 µL of 2.5 mM sodium acetate to each well. A multimode plate reader (Multiskan™ GO, Thermo Scientific, Waltham, MA, USA) was used to measure fluorescence with an excitation wavelength of 545 nm and an emission wavelength of 585 nm. The IC_50_ value was determined by plotting the relative fluorescence units per hour (RFU/h) against the logarithmic inhibitor concentrations. All experiments were conducted in triplicate.

### 4.5. Tau Inhibition Activity Assay

#### 4.5.1. Tissue Culture of Adherent Cells

SH-SY5Y cells were grown in a complete growth medium comprising Dulbecco’s Modified Eagle Medium (DMEM) supplemented with 10% fetal bovine serum (FBS) and 1% penicillin/streptomycin. For treatment, adherent SH-SY5Y cells were plated in 96-well plates at a density of 2 × 10^6^ cells per well. Levosimendan (Sigma, St. Louis, MO, USA) was used as an inhibitor to verify the effect on tau inhibition. Upon treatment with 20 μM of each substance, the cells were cultured for 24 h. The supernatant was then collected and used in subsequent ELISA assays.

#### 4.5.2. Tau ELISA

The tau levels were assessed using a Human Tau ELISA kit (Abcam, Cambridge, UK). After treatment, the supernatant was diluted five times. The diluted supernatant, capture antibody, and detection antibody were mixed in a 2:1:1 ratio and incubated for 1 h. The wells were then washed three times with a wash buffer. After removing the buffer, 100 µL of 3,3′,5,5′-tetramethylbenzidine developer was added and incubated for 10 min, followed by the addition of 100 µL stop solution. The optical density was measured at 450 nm. Data analysis was performed by calculating the mean and standard deviation using GraphPad Prism 8 (GraphPad, San Diego, CA, USA).

### 4.6. Cytotoxicity Assay

#### 4.6.1. Cell Culture

The murine macrophage cell line RAW 264.7 (KCLB number 40071; Korean Cell Line Bank, Seoul, Republic of Korea) was cultured in DMEM (Sigma-Aldrich, St. Louis, MO, USA) supplemented with 10% heat-inactivated FBS (Invitrogen, Burlington, ON, Canada) and 1% (*w*/*v*) antibiotic–antimycotic solution (Invitrogen, Grand Island, NY, USA) under 5% CO_2_ at 37 °C.

#### 4.6.2. MTT Assay

The cytotoxicity of the cells was evaluated using an MTT colorimetric assay (3-(4,5)-dimethylthiazol-2-yl-2,5-diphenyltetrazolium bromide, Amresco, Solon, OH, USA). RAW 264.7 cells were seeded at a concentration of 2 × 10^5^ cells/mL in 96-well plates and treated with various concentrations of Ramalin and its derivatives for 24 h. Following treatment, 5 µL of MTT solution (5 mg/mL in PBS) was added to each well, and the cells were incubated for 4 h at 37 °C. After incubation, 100 µL of fresh DMSO was added to dissolve the formazan crystals, and the cells were incubated for 10 min. Absorbance at 570 nm was then measured using a microplate reader (Thermo Scientific Inc., San Diego, CA, USA). Relative cell viability was calculated by comparing the absorbance values to those of the untreated control group. All assays were performed in triplicate.

### 4.7. NO Assay and NLRP3 ELISA

#### 4.7.1. Determination of Nitric Oxide Production

Nitrite accumulation served as an indicator of NO production in the medium, and the nitrite concentration was assessed by analyzing the culture supernatants using the Griess reagent (1% sulfanilamide, 0.1% N-(1-naphthyl)-ethylenediamine dihydrochloride, and 5% phosphoric acid). To quantify the nitrite levels, 1 × 10^6^ cells/mL were seeded in 96-well plates, followed by treatment with the specified concentrations of Ramalin and its derivatives at 37 °C for 1 h. Afterward, the cells were stimulated with 0.5 μg/mL of lipopolysaccharide (LPS, 0.5 μg/mL, Sigma-Aldrich, St. Louis, MO, USA) for 24 h in a final volume of 200 μL. Subsequently, 100 μL of the cell culture supernatants were combined with 100 μL of Griess reagent in a 96-well plate. A standard curve was prepared using sodium nitrite, and the nitrite concentration was determined by measuring the absorbance at 540 nm using a microplate reader. All experiments were carried out in triplicate.

#### 4.7.2. NLRP3 ELISA

RAW 264.7 macrophages were seeded at a density of 5 × 10^5^ cells/well in 96-well plates and treated with 20 µM of RA-Hyd-Me, RA-Hyd-Me-tol, RA-Benzo, and RA-Sali for 1 h and then stimulated with 0.5 μg/mL LPS for 24 h. Culture supernatant levels of NOD-like receptor pyrine domain containing-3 (NLRP3) were determined using a commercially available ELISA kit (Lsbio, Seattle, WA, USA) and following the instructions of the manufacturer. The absorbance of the plate was read at a wavelength of 450 nm. NLRP3 was determined from a standard curve. The concentrations were expressed as ng/mL. To obtain statistical significance, all experiments were carried out in duplicate. 

### 4.8. Mini Ames Test

The mutagenic potential of the test substances was evaluated using a microplate Ames assay (Xenometrics MPF^TM^ Ames Assay, AG, Allschwil, Switzerland), which employs the bacterial strains Salmonella typhimurium TA98 and TA100. The assay was conducted according to the xenometric calculation method, a validated approach for determining the mutagenicity of chemical compounds. The AMES MPF Penta 2 mutagenicity assay kit was utilized for this study. The bacterial strains were cultured in a growth medium supplemented with 1% (*v*/*v*) penicillin/streptomycin at 37 °C, with shaking at 250 rpm for 12 to 16 h. The test substances were prepared by serially diluting them into six different concentration levels. Frozen bacterial strains were pre-cultured under the conditions described above, and the bacterial suspensions were exposed to the test substances, negative control (DMSO), and positive controls in the presence or absence of the metabolic activation system (S9) for 90 min at 37 °C. For the positive controls, the TA98 strain was exposed to 2-nitrofluorene (2-NF) without S9 and 2-aminoanthracene (2-AA) with S9, while the TA100 strain was exposed to 4-nitroquinoline 1-oxide (4-NQO) without S9 and 2-AA with S9. After the exposure period, the mixtures were combined with the indicator medium and distributed into 384-well plates, which were incubated at 37 °C for 24 to 48 h. The mutagenic potential was assessed by counting the number of revertant colonies in each well. Positive results were defined as a two-fold or greater increase in the number of revertant colonies compared to the negative control or as a dose-dependent increase in revertant colonies.

### 4.9. Acetylcholinesterase Activity Assay

#### AChE Assay

The AChE inhibitory activity was assayed following an adaptation of the spectrophotometric method reported by Ellman et al. [41]. The Acetylcholinesterase Inhibitor Screening Kit (catalog number MAK324) and purified AChE (catalog number C3389) were purchased from Sigma-Aldrich. Enzyme solutions were prepared by dissolving lyophilized powder in double-distilled water. The AChE inhibitory activity was measured using a clear 96-well flat-bottom plate. The absorbance was read on a fluorescence spectrometer (Thermo Scientific Inc., San Diego, CA, USA) in duplicate experiments with two control wells: a standard (no enzyme) well and one well containing the AChE reference enzyme (no-inhibitor control). The experimental procedures for AChE activity assays were performed according to the technical bulletins of the acetylcholinesterase activity assay kit (MAK324; Sigma-Aldrich). Purified AChE was prepared to a concentration of 500 units/mL. A reaction mix for each well was prepared by mixing the following into a clean tube: 154 μL of assay buffer (catalog number MAK324A), 1 μL of substrate (100 mM, catalog number MAK324B), and 0.5 μL of 5,5′-dithiobis(nitrobenzoic acid) (DNTB, catalog number MAK324C). The reaction was initiated by the addition of 45 μL of the assay buffer, 5 μL of the enzyme, and the investigated compounds (5 μL) to the wells to obtain the final concentrations of 6.25, 12.5, 25, and 50 μM. A positive control of donepezil was used in the same range of concentrations. The plate was incubated for 15 min. The reaction mix (150 μL) was then added to each sample, the control (no enzyme), and the no-inhibitor control wells. The plate was tapped to mix. Absorbance was measured at 412 nm at 0 min and then at 10 min. The AChE inhibitory activity was calculated as the % of inhibition. All samples were assayed in triplicate.

### 4.10. Statistical Analysis

Graphs and statistical analyses were created using GraphPad Prism 8. Data were analyzed using a one-way analysis of variance (ANOVA), followed by Dunnett’s multiple comparison test. The results are expressed as the mean ± standard deviation of three independent experiments, with statistical significance levels set at * *p* < 0.05, ** *p* < 0.01, and *** *p* < 0.005.

### 4.11. Use of AI-Assisted Tools

In the preparation of this manuscript, ChatGPT-4, an AI language model developed by OpenAI, was used to assist in various tasks including translation, proofreading for typographical errors, and contextual review to ensure clarity and accuracy of the content. The AI tool was employed as a supplementary aid to improve the quality of the writing, and all outputs generated by the AI were carefully reviewed and edited by the authors to ensure the originality, validity, and integrity of the manuscript. The use of ChatGPT-4 does not meet the criteria for authorship, in accordance with MDPI’s guidelines, and was therefore not included as an author.

## 5. Conclusions

In this study, four novel Ramalin derivatives (RA-Hyd-Me, RA-Hyd-Me-Tol, RA-Sali, and RA-Benzo) were prepared and their stabilities and therapeutic potentials against Alzheimer’s disease (AD) were evaluated. The synthesized derivatives showed significantly improved stabilities compared to that of the parent compound, Ramalin, especially at elevated temperatures, thereby overcoming a major challenge in the development of reliable AD treatments. The incorporation of functional groups, such as methyl and benzamido groups, was found to play a crucial role in preventing oxidative degradation, thereby stabilizing the derivatized compounds. Despite a reduction in their antioxidant activities, particularly in the cases of RA-Benzo and RA-Sali, the prepared derivatives exhibited promising multi-target therapeutic properties. More specifically, RA-Sali demonstrated the most potent BACE-1 inhibitory activity, whilst significantly inhibiting tau phosphorylation, rendering it a promising candidate for further development. However, the obtained derivatives did not exhibit inhibitory activity against acetylcholinesterase, indicating that although these compounds have potential as multi-target therapeutic agents, they may not address all the pathways involved in AD. Additionally, the low molecular weights and favorable physicochemical properties of these derivatives indicate their potential ability to cross the blood–brain barrier (BBB) and exert therapeutic effects directly in the brain. However, further optimization is required to ensure effective permeability of the BBB, and to enhance their efficacies, particularly in terms of their antioxidant and anti-inflammatory activities. Thus, although the synthesized Ramalin derivatives (particularly RA-Sali) show promise as potential therapeutic agents for AD, further structural refinements and studies are required to fully understand their potential. This study represents a significant advancement in the development of stable and effective AD treatments and provides a foundation for future therapeutic strategies.

## Data Availability

The data presented in this study are available in this article.

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
