# Peer review of "Therapeutic Potential of Ramalin Derivatives with Enhanced Stability in the Treatment of Alzheimer’s Disease"

_molecules, 2024, doi:10.3390/molecules29225223_

Round 1
Reviewer 1 Report
Comments and Suggestions for Authors
In manuscript “Therapeutic Potential of Ramalin Derivatives with Enhanced 2 Stability in the Treatment of Alzheimer’s Disease,” authors describe the synthesis of 4 derivatives of Ramalin. A biological evaluation study, which included Antioxidant activity, BACE-1 Inhibitory Activity, ELISA, anti-inflammatory activity, acetylcholinesterase inhibitory activity, mutagenic potential and toxicity was also presented. I recommend the publication of this manuscript in Molecules after a major revision:
-Authors must thoroughly revise the oxidation mechanism presented in scheme 2, due to this mechanism in its current form is riddled with errors. In this regard, the following references can help to propose a more logic mechanism: (a) Journal of Hazardous Materials, 2020, 387, 122000 (b) International Journal of Environmental Science and Technology, 2023, 20, 3901–3909.
-In scheme 1, Authors should revise the structure of p-Glu-anhydride (an oxygen atom is missing!!). As recommendation, author authors should provide a proper numbering of compounds rather than identifying them by name/abbreviation. The numbering of compounds should be clearly indicated in both in schemes and in the main text.
-Authors should revise the use of acronyms to refer to chemicals, for example, AcOEt is more appropriate and common than EA for ethyl acetate.
-Authors should include the IR spectra characterization for all compounds.
-Authors should check thoroughly typographical mistakes, for example:
-Page 6, line 196, “Inhibotory.”
Author Response
In manuscript “Therapeutic Potential of Ramalin Derivatives with Enhanced 2 Stability in the Treatment of Alzheimer’s Disease,” authors describe the synthesis of 4 derivatives of Ramalin. A biological evaluation study, which included Antioxidant activity, BACE-1 Inhibitory Activity, ELISA, anti-inflammatory activity, acetylcholinesterase inhibitory activity, mutagenic potential and toxicity was also presented. I recommend the publication of this manuscript in Molecules after a major revision:
Comment 1: Authors must thoroughly revise the oxidation mechanism presented in scheme 2, due to this mechanism in its current form is riddled with errors. In this regard, the following references can help to propose a more logic mechanism: (a) Journal of Hazardous Materials, 2020, 387, 122000 (b) International Journal of Environmental Science and Technology, 2023, 20, 3901–3909.
Response 1
We sincerely thank the reviewer for pointing out the errors in the oxidation mechanism presented in Scheme 2. We acknowledge the issues with the previous mechanism and appreciate the valuable feedback. Upon careful consideration, we realized that attempting to present the mechanism based solely on predictions was indeed problematic. As a result, we have thoroughly revised Scheme 2, making significant modifications to the illustration. Additionally, the related text has been deleted and rewritten to ensure clarity and accuracy.
In the revised version, we have provided concrete evidence supporting the decomposition of Ramalin by oxygen and explained our approach to structural modification to prevent this degradation. We believe these revisions have strengthened the overall presentation of the oxidation process and the proposed solution to improve Ramalin's stability.
We hope this addresses the reviewer's concerns, and we appreciate your feedback, which has greatly improved the quality of our manuscript.
Comment 2: In scheme 1, Authors should revise the structure of p-Glu-anhydride (an oxygen atom is missing!!). As recommendation, author authors should provide a proper numbering of compounds rather than identifying them by name/abbreviation. The numbering of compounds should be clearly indicated in both in schemes and in the main text.
Response 2
We have identified and corrected the error in the structure of p-Glu-anhydride in Scheme 2. Similarly, the same issue was found in Scheme 1, and we have made the necessary corrections there as well. We appreciate your suggestion to label the compounds with numbers, and we agree that it is a commonly used practice. However, since this scheme represents a general synthetic process of derivatives rather than precise structures, we chose to use structural names instead of numbering. Therefore, we have not implemented the numbering system in this case. Once again, we sincerely thank you for your valuable suggestion.
Comment 3: Authors should revise the use of acronyms to refer to chemicals, for example, AcOEt is more appropriate and common than EA for ethyl acetate.
Response 3
As per the reviewer's suggestion, we have revised the use of "EA" to "AcOEt" for ethyl acetate throughout the manuscript. Other acronyms, apart from "EA," have been retained. We appreciate your valuable feedback on this matter.
Comment 4: Authors should include the IR spectra characterization for all compounds.
Response 4
We have completed the FT-IR analysis for the four synthesized Ramalin derivatives and have included the results in the Supporting Information. Thank you for your suggestion.
Comment 5: Authors should check thoroughly typographical mistakes, for example: Page 6, line 196, “Inhibotory.”
Response 5
We have identified and corrected the typographical error “Inhibotory” to “Inhibitory” on page 6, line 196. Additionally, we have thoroughly reviewed the manuscript for any further typographical errors. Thank you for your feedback.
We sincerely thank the reviewer for their thoughtful and constructive feedback on our manuscript titled “Therapeutic Potential of Ramalin Derivatives with Enhanced Stability in the Treatment of Alzheimer’s Disease.” We have addressed all the suggested revisions, including correcting the typographical errors, revising the oxidation mechanism in Scheme 2, updating the use of chemical acronyms, and adding the requested FT-IR analysis of the synthesized Ramalin derivatives to the Supporting Information. We believe that these revisions have significantly improved the quality of our manuscript.
We greatly appreciate the reviewer’s recommendation for publication in Molecules and look forward to the opportunity to contribute to the scientific community through this work. Thank you once again for your invaluable feedback and consideration.
Reviewer 2 Report
Comments and Suggestions for Authors
This manuscript describes the synthesis and biological activity of new ramalin derivatives (4 derivatives) with promising anti-AD activity.
The manuscript is sound and interesting, but it requires some improvements:
- page 6, line 173 - "It was found that their 173 antioxidant activities were significantly lower than that of Ramalin itself" - this is not true for all the derivatives and therefore the sentence should be reconsidered.
- Table 2 - in the DPPH IC50 column, some numbers are incorrectly separated by a comma.
Author Response
This manuscript describes the synthesis and biological activity of new ramalin derivatives (4 derivatives) with promising anti-AD activity.
The manuscript is sound and interesting, but it requires some improvements:
Comment 1: page 6, line 173 - "It was found that their 173 antioxidant activities were significantly lower than that of Ramalin itself" - this is not true for all the derivatives and therefore the sentence should be reconsidered.
Comment 2: Table 2 - in the DPPH IC50 column, some numbers are incorrectly separated by a comma.
Response 1 & 2
We sincerely thank the reviewer for their insightful comments and suggestions to improve our manuscript. We have carefully addressed the points raised as follows:
- Comment on Table 2 (DPPH IC50 column): After reviewing the reviewer’s suggestion regarding the DPPH IC50 values, we have re-examined the table and confirmed that the values for RA-Benzo and RA-Sali were correctly presented in micromolar (µM) units as intended. Specifically, the DPPH IC50 values are RA-Benzo = 3,862 µM (3.86 mM) and RA-Sali = 1,052 µM (1.05 mM). We appreciate the reviewer’s attention to detail, but no further changes were necessary as the correct formatting was already in place.
- Comment on page 6, line 173: Upon further review of the data, we believe that the original statement regarding the antioxidant activities is accurate for the mentioned derivatives, as their antioxidant effects are indeed much lower. Therefore, we do not see the need for any revisions to the text on page 6, line 173.
We appreciate your thorough review and helpful feedback, which has allowed us to confirm the accuracy of the manuscript. We believe the revised version continues to reflect our findings correctly.
Thank you again for your valuable comments and consideration.
Reviewer 3 Report
Comments and Suggestions for Authors
This paper describes a structure-activity relationship study of a natural product, Ramalin, that has been proposed as a lead compound for the treatment of Alzheimer’s disease. The natural product has an unusual hydrazine-containing structure. The lead compound has some desirable characteristics (e.g. antioxidant activity) but also some undesirable characteristics (e.g. instability, mutagenicity, poor bioavailability). In this paper, four new analogues are synthesised with modifications to the hydrazine moiety. The new analogues all have poorer antioxidant activity, suggesting that it is the hydrazine moiety that is primarily responsible for antioxidant activity. But the new analogues do show improvements in other respects e.g. stability, BACE-1 inhibitory activity, and lower mutagenicity. BBB permeability was not tested directly but was predicted using a chemoinformatics approach, and this suggested that most compounds are likely to be BBB permeable except, frustratingly, the best analogue (which has a salicylic acid moiety). Developing a treatment for AD requires simultaneous optimisation of several independent parameters, and this paper is a good illustration of the difficulty of that task.
I believe that the paper is scientificially sound and deserves to be published in Molecules. My most significant criticism concerns the proposed mechanisms of degradation and antioxidant activity, which I believe are flawed; I have suggested alternatives (see below). My other comments just relate to the presentation style.
· There are several problems with the oxidation process proposed in Figure 2:
o The resonance form shown at the top right appears impossible to me, because the oxygen seems to have a share of 10 valence electrons (double bond to carbon, single bond to hydrogen, and two lone pairs).
o The curly arrows from the top middle structure do not correctly lead to the top right structure.
o Bottom right structure, the NH group is “upside down”.
o The dehydration reaction doesn’t look right, because it involves electrons moving away from a positively charged centre towards a negatively charged centre.
o Another problem with the dehydration reaction is that it doesn’t depict where the ring hydrogen comes from.
· Might it make more sense to invoke a base-promoted deprotonation of the NH adjacent to the aryl group, pushing electrons in to make a N=N double bond, migration of the second NH hydrogen to the benzene ring via a 5-membered transition state, displacing the phenol OH (which might have been protonated prior to give water as the leaving group)? This would be a simple acid / base promoted reaction, which would account for the degradation of the compound, separate from antioxidant activity which might simply involve oxidation of the NH-NH group to N=N. For this reason, I find the frequent description of “electron flow” between the hydrazine and the phenol in the context of antioxidant activity to be unconvincing.
· A citation needs to provided for the statement “as confirmed by mass spectral measurements”.
· In all chemical structure drawings, please indicate the stereochemistry
· For all schemes / tables / figures, it would be better to use compound numbers rather than abbreviated names.
· Section 2.3: the discussion is a bit repetitive and obvious, e.g. when quoting a higher IC50 value it is not necessary to immediately explain that this means weaker binding, or when describing the effect of moving a methyl group it is not very illuminating to state that the methyl group must be having an interaction with the BACE-1 active site. I suggest to make this paragraph more concise. Also, in cases where two compounds have IC50 values with overlapping ranges, be careful to avoid a definitive statement that one is more potent than the other.
· There are other examples where non-illuminating “discussion” is provided in the results section, e.g. final paragraph of section 2.8. I suggest to make the entire results section more concise by removing such material.
· Materials and methods: for synthetic procedures, it is traditional to omit any mention of glassware (e.g. round bottom flasks) or equipment (e.g. rotary evaporator).
· In the supporting info, it would be helpful to superimpose a structure drawing on each spectrum.
Author Response
This paper describes a structure-activity relationship study of a natural product, Ramalin, that has been proposed as a lead compound for the treatment of Alzheimer’s disease. The natural product has an unusual hydrazine-containing structure. The lead compound has some desirable characteristics (e.g. antioxidant activity) but also some undesirable characteristics (e.g. instability, mutagenicity, poor bioavailability). In this paper, four new analogues are synthesised with modifications to the hydrazine moiety. The new analogues all have poorer antioxidant activity, suggesting that it is the hydrazine moiety that is primarily responsible for antioxidant activity. But the new analogues do show improvements in other respects e.g. stability, BACE-1 inhibitory activity, and lower mutagenicity. BBB permeability was not tested directly but was predicted using a chemoinformatics approach, and this suggested that most compounds are likely to be BBB permeable except, frustratingly, the best analogue (which has a salicylic acid moiety). Developing a treatment for AD requires simultaneous optimisation of several independent parameters, and this paper is a good illustration of the difficulty of that task.
I believe that the paper is scientificially sound and deserves to be published in Molecules. My most significant criticism concerns the proposed mechanisms of degradation and antioxidant activity, which I believe are flawed; I have suggested alternatives (see below). My other comments just relate to the presentation style.
Comment 1: There are several problems with the oxidation process proposed in Figure 2:
- The resonance form shown at the top right appears impossible to me, because the oxygen seems to have a share of 10 valence electrons (double bond to carbon, single bond to hydrogen, and two lone pairs).
- The curly arrows from the top middle structure do not correctly lead to the top right structure.
- Bottom right structure, the NH group is “upside down”.
- The dehydration reaction doesn’t look right, because it involves electrons moving away from a positively charged centre towards a negatively charged centre.
- Another problem with the dehydration reaction is that it doesn’t depict where the ring hydrogen comes from.
Might it make more sense to invoke a base-promoted deprotonation of the NH adjacent to the aryl group, pushing electrons in to make a N=N double bond, migration of the second NH hydrogen to the benzene ring via a 5-membered transition state, displacing the phenol OH (which might have been protonated prior to give water as the leaving group)? This would be a simple acid / base promoted reaction, which would account for the degradation of the compound, separate from antioxidant activity which might simply involve oxidation of the NH-NH group to N=N. For this reason, I find the frequent description of “electron flow” between the hydrazine and the phenol in the context of antioxidant activity to be unconvincing.
Response 1
We acknowledge the errors in the oxidation mechanism proposed in Figure 2, as pointed out by the reviewer. We agree that relying solely on predictions rather than experimental evidence has led to inaccuracies in the depiction of the mechanism. As a result, we have revised the figure to correct these issues.
Based on evidence of Ramalin’s oxidation by oxygen, we have analyzed the mass values and predicted the formation of the RA-Azo structure. Although attempts were made to isolate and purify RA-Azo, it decomposed too rapidly, making purification impossible. In light of this, we have designed derivatives to prevent the transformation into RA-Azo, aiming to improve the stability of Ramalin. These revisions have been incorporated into the manuscript, and the figure has been updated accordingly.
Comment 2: A citation needs to provided for the statement “as confirmed by mass spectral measurements”.
Response 2
We have included the relevant data to support the statement “as confirmed by mass spectral measurements” in Supporting Information S33.
Thank you for your suggestion.
Comment 3: In all chemical structure drawings, please indicate the stereochemistry
Response 3
Thank you for your suggestion. We have indicated the stereochemistry in all chemical structure drawings.
Comment 4: For all schemes / tables / figures, it would be better to use compound numbers rather than abbreviated names.
Response 4
We fully agree with the reviewer's suggestion to use compound numbers rather than abbreviated names. However, as there are relatively few compounds and the schemes depict a general synthesis method, we opted to use compound names for clarity. We kindly ask for your understanding in this choice, though we are open to adjusting the notation should the reviewer have further recommendations.
Thank you for your valuable feedback.
Comment 5: Section 2.3: the discussion is a bit repetitive and obvious, e.g. when quoting a higher IC50 value it is not necessary to immediately explain that this means weaker binding, or when describing the effect of moving a methyl group it is not very illuminating to state that the methyl group must be having an interaction with the BACE-1 active site. I suggest to make this paragraph more concise. Also, in cases where two compounds have IC50 values with overlapping ranges, be careful to avoid a definitive statement that one is more potent than the other.
Response 5
We acknowledge the reviewer’s feedback regarding the repetitive and obvious discussions in Section 2.3. We have revised the section to make it more concise, avoiding unnecessary explanations and focusing on key findings. Additionally, we have ensured that no definitive statements are made regarding the potency of compounds with overlapping IC50 ranges.
By focusing on the essential findings and avoiding overly descriptive details, the revised section now addresses the reviewer’s concerns. Thank you for your valuable feedback, which has improved the clarity and conciseness of the manuscript.
Comment 6: There are other examples where non-illuminating “discussion” is provided in the results section, e.g. final paragraph of section 2.8. I suggest to make the entire results section more concise by removing such material.
Response 6
We appreciate the reviewer’s suggestion to make the results section more concise by removing non-illuminating discussion. In response, we have revised the final paragraph of section 2.8 to streamline the content. The updated version focuses solely on the key findings, specifically stating that Ramalin and its derivatives have minimal AChE inhibitory activity, which suggests limited potential as AChE inhibitors. Unnecessary elaboration on further research and structural modifications has been removed to enhance clarity and conciseness.
Thank you for your valuable feedback, which has helped improve the overall presentation of the results.
Comment 7: Materials and methods: for synthetic procedures, it is traditional to omit any mention of glassware (e.g. round bottom flasks) or equipment (e.g. rotary evaporator).
Response 7
As per the reviewer’s suggestion, we have revised the "Materials and Methods" section by omitting unnecessary mentions of glassware (e.g., round bottom flasks) and equipment (e.g., rotary evaporator). The section has been rewritten to focus on the essential details of the synthetic procedures.
Thank you for your valuable feedback
Comment 8: In the supporting info, it would be helpful to superimpose a structure drawing on each spectrum.
Response 8
As per the reviewer’s suggestion, we have added structure drawings to the 1H, 13C NMR data and HRMS data in the Supporting Information to enhance clarity.
Thank you for your valuable feedback
We sincerely thank the reviewer for their thorough and constructive feedback, which has greatly improved the clarity and quality of our manuscript. Based on your suggestions, we have made several revisions, including correcting the oxidation mechanism, streamlining the discussion in the results section, and improving the "Materials and Methods" section by removing unnecessary details. Additionally, we have added structural drawings to the supporting information as per your recommendation.
We believe that these revisions have strengthened the manuscript and enhanced its overall presentation. We deeply appreciate your valuable insights and suggestions, which have been instrumental in refining our work. Thank you again for your careful review and thoughtful comments.
Round 2
Reviewer 1 Report
Comments and Suggestions for Authors
After revision of revised manuscript, authors have made suggested changes and now manuscript us suitable for publication in its current form.